# Novel Mesogenic Vinyl Ketone Monomers and Their Based Polymers

**DOI:** 10.3390/polym15010005

**Published:** 2022-12-20

**Authors:** Yaroslav I. Derikov, Daniil R. Belousov, Alexander V. Finko, Georgii A. Shandryuk, Nina M. Kuz’menok, Sergei G. Mikhalyonok, Vladimir S. Bezborodov, Elena V. Chernikova, Raisa V. Talroze

**Affiliations:** 1A.V. Topchiev Institute of Petrochemical Synthesis Russian Academy of Sciences, 119991 Moscow, Russia; 2Faculty of Petroleum Chemistry and Polymeric Materials, D. Mendeleev University of Chemical Technology of Russia, 125047 Moscow, Russia; 3Faculty of Chemistry, M.V. Lomonosov Moscow State University, 119991 Moscow, Russia; 4Department of Chemistry, Belarusian State Technological University, 220006 Minsk, Belarus

**Keywords:** vinyl ketones, liquid crystal polymers, RAFT, photodegradation

## Abstract

In the present research, we have synthesized new vinyl ketone monomers with mesogenic substituents, namely, 8-(3′-chloro-4′-pentyl-[1,1′-biphenyl-4-oxy)oct-1-en-3-one (**BVK**) and 8-[2′-chloro-4‴-octyl-[1,1′:4′,1″:4″,1‴-quaterphenyl-4-oxy]oct-1-en-3-one (**QVK**). The comparison of **BVK**, **QVK**, and previously synthesized 8-((4″-((*1R,4S*)-4-butylcyclohexyl)-2′-chloro-[1,1′,4′,1″-terphenyl]-4-yl)oxy)oct-1-en-3-one (**TVK**) has revealed that all of them are able to form crystals, while their ability to exhibit liquid crystalline behavior depends on the number of phenyl substituents attached to the para-position of the phenoxy group and is observed for **TVK** and **QVK** only. All of the monomers are able to achieve self-polymerization upon heating and free radical polymerization in bulk or in solution under the action of the common radical initiator AIBN. We have also succeeded in the RAFT polymerization of the synthesized vinyl ketones **BVK** and **TVK** using asymmetrical trithiocarbonates. The synthesized poly(vinyl ketones) exhibit LC behavior and are able to undergo photodegradation upon UV irradiation.

## 1. Introduction

Vinyl ketones CH_2_=CH–C(=O)–R belongs to an interesting class of monomers as some representatives are able to polymerize through radical, cationic, and anionic mechanisms [1]. Most vinyl ketones are readily polymerized via a radical mechanism due to the stabilization of the produced radical active center. Besides, methyl- [2], t-butyl- [3], phenyl vinyl ketones [4], some higher alkyl vinyl ketones [3], and optically active vinyl ketones, such as [(s)-1-methyl-propyl]vinyl ketone, [(s)-2-methyl-butyl]vinyl ketone, and [(s)-3-methyl-pentyl]vinyl ketone [5] polymerize in bulk even at room temperature. Azobisisobutyronitrile (AIBN) is found to be a more efficient radical initiator than benzoyl peroxide in the bulk polymerization of methyl isopropenyl ketone [6], while both initiators are effective in the radical polymerization of phenyl vinyl ketone in hydrocarbon solution [7,8]. Both ultraviolet and gamma radiation initiate radical polymerization of methyl vinyl ketone [9], methyl isopropenyl ketone [10,11,12], and phenyl vinyl ketone [4] providing the formation of cross-linked polymers in bulk or in acetone and a colored polymer with intramolecular aldol linkages:



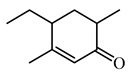



A series of copolymers of acrylonitrile with vinyl ketones, including *t*-butyl-, phenyl-, isopropyl-, and ethyl vinyl ketone are described in [13]. The reactivity of the vinyl ketone monomers depends on the polar nature of the substituent but it is not affected by its bulkiness.

The growing interest in vinyl ketone-based polymers started in the mid-1940s and continues to this day, and is caused by their high potential in various applications. Polyvinyl ketones (PVKs) are photo-responsive polymers with high photo degradability [14,15,16,17,18]. Due to their photo-responsive nature, PVKs have been used as functional and green materials in agriculture [19], imaging and microfabrication technologies [20], optical sensing [21], nanoporous materials production [22,23], and as carbon fiber precursors [24]. Although PVKs have been known for a long time [25,26], the main interest was focused on vinyl ketones with relatively small substituents, such as methyl, butyl, and phenyl [1]. These particular monomers have been applied first in reversible deactivation radical polymerization (RDRP) [27,28,29,30,31,32]. RDRP is a versatile technique that provides the synthesis of well-defined polymers with desirable architecture [33,34,35]. Among known RDRP techniques, only reversible addition–fragmentation chain transfer (RAFT) polymerization has succeeded in the control synthesis of PVKs [30,31,32]. The first example of RAFT polymerization of methyl- and phenyl vinyl ketones using (S)-1-dodecyl-(S′)-(α,α′-dimethyl-α′′-acetic acid)trithiocarbonate as a RAFT agent is described in [30]. The living nature of the process was confirmed by the formation of PVKs with controlled M_n_, dispersity Đ = M_w_/M_n_ below 1.2, and by chain extension with styrene. Further RAFT technique was applied to the synthesis of the photo-degradable triblock copolymers polystyrene-*b*-poly(phenyl vinyl ketone)-*b*-polymethyl acrylate and amphiphilic polystyrene-*b*-poly(phenyl vinyl ketone)-*b*-polyacrylic acid [31]. The latter is able to self-assemble into spherical micelles in water and can be entirely disintegrated under UV irradiation due to photocleavage of the PVK block. Photo-induced electron transfer (PET) RAFT polymerization can also be used to polymerize vinyl ketones, such as methyl-, ethyl-, and phenyl vinyl ketones [32,35]. All these examples illustrate that well-defined poly(vinyl ketones) derivatives can be prepared by RAFT technique in the range of M_n_ from 2 to 40 kg·mol^−1^.

Significantly less attention is paid to PVKs with bulky substituents [36,37,38]. As an example, isopropenyl 4-[bis(4-methylphenyl)amino]phenyl ketone was homo- and copolymerized by radical mechanism [36]. The obtained polymer (Figure 1) possesses a rather rigid, restricted conformation in which the side-chain chromophores are π-stacked, and they reveal hypochromicity and redshifts in absorbance spectra and redshifts in emission spectra compared with the monomeric unit model. 

A further example includes an adamantyl vinyl ketone which is able to participate in RAFT polymerization without added initiator and can form both random and block copolymers with methyl methacrylate [37]. Poly(adamantyl vinyl ketone) (Figure 2) undergoes photodegradation after exposure to UV light, which results in a decrease in polymer molecular weight and the formation of a new polymer and adamantane as the major products. 

Finally, a new vinyl ketone monomer containing a bulky terphenyl group (Figure 3) has been described in our previous publication [38]. A calorimetric study has revealed that the formation of oligomer with M_n_ about 4 kg·mol^−1^ and a yield of more than 70% occurs upon heating TVK monomer in the DSC cell above 100 °C. Thus, it may be supposed that TVK undergoes self-polymerization at elevated temperatures. However, the properties of the synthesized oligomer have not been studied.

In the present work, we have developed two similar vinyl ketones differing by the structure of the substituent attached to the phenoxy group (Figure 4) and have tested different ways of the radical polymerization of **TVK** and the newly synthesized vinyl ketones. The bulky substituent in all three monomers may be considered a mesogenic group and, hence, can provide liquid crystalline properties of both monomers and their based polymers.

## 2. Experimental

### 2.1. Materials and Methods

All starting materials and solvents were supplied by Alfa Aesar, Sigma-Aldrich, and used for the preparation of **BVK** and **QVK**. Radical initiator AIBN was recrystallized from ethanol and stored in the freezer before use. RAFT-agents, namely 2-(dodecylthiocarbonothioylthio)propionic acid (DoPAT) and 2-cyano-2-propyl dodecyl trihtiocarbonate (CPDTC) were supplied by Sigma-Aldrich and used without purification. All the solvents used in the synthesis and polymer purification and characterization, namely chloroform, ethyl alcohol, toluene, hexane, acetone, 1,4-dioxane, and tetrahydrofuran (THF) were preliminarily distilled. 

### 2.2. Monomer Synthesis

**TVK** was synthesized and characterized according to the procedure described elsewhere [39]. 

The synthesis of **BVK** and **QVK** was performed according to Figure 1 and Figure 2, respectively. Synthetic methods for both monomers were identical as described in [39,40]. 

Briefly, the mixture of 3′-chloro-4′-pentyl-[1,1′-biphenyl]-4-ethoxy (**1**) or 2′-chloro-4‴-octyl-4-methoxy [1,1′:4′,1″:4″,1‴-quaterphenyl] (**1a**) with acetic acid and 59% hydroiodic acid was refluxed for 12 h, extracted with methylene chloride, washed by the diluted solution of sodium thiosulphate and water, and dried over magnesium sulphate. Thereafter, 3′-chloro-4′-pentyl-[1,1′-biphenyl]-4-ol (**2**) or 2′-chloro-4‴-octyl-[1,1′:4′,1″:4″,1‴-quaterphenyl]-4-ol (**2a**) was recrystallized from toluene–heptane mixture. Compounds (**2**) or (**2a**) were mixed with ethyl 6-bromohexanoate, K_2_CO_3_, and KI in methylethyl ketone. The mixture was refluxed for 22 h, and the product was extracted with methylene chloride, washed with water, and dried over magnesium sulfate. Thereafter, 6-[3′-chloro-4′-pentyl-[1,1′-biphenyl]-4-yl)-oxy)hexanoate (**3**) or ethyl 6-[2′-chloro-4‴-octyl-[1,1′:4′,1″:4″,1‴-quaterphenyl]-4-yl)-oxy)hexanoate (**3a**) was recrystallized from ethyl acetate. Ethyl bromide in absolute diethyl ether was added to magnesium turnings with stirring. Compounds (**3**) or (**3a**) in absolute diethyl ether and tetraisopropyl orthotitanate were added, followed by slow addition of the freshly prepared ethylmagnesium bromide at room temperature upon stirring. After 2 h, 10% sulfuric acid was added to the reaction mixture at room temperature. The ether layer was separated and the aqueous part was extracted with ether. The ether extracts were dried over MgSO_4_ and separated by chromatography on silica gel. Thereafter, 1-(5-([3′-Chloro-4′-pentyl-[1,1′-biphenyl]-4-yl)-oxy)pentyl)cyclopropan-1-ol (**4**) or 1-(5-[2′-chloro-4‴-octyl-[1,1′:4′,1″:4″,1‴-quaterphenyl]-4-yl)-oxy)pentyl)cyclopro-pan-1-ol (**4a**) were crystallized from isopropyl alcohol. Methanol solution of bromine was slowly added to a methanol:water (60:1) solution of compound (**4**) or (**4a**) upon stirring. After 10 min, water was added and the precipitate was filtered off, washed with water, and dissolved in CCl_4_. A Triethylamine solution of hydroquinone was added first, and 5% sulfuric acid was added after stirring at 80–90 °C and cooling. The organic layer was separated, the aqueous part was extracted with benzene, and the extract was separated by chromatography on silica gel. The **BVK** and **QVK** were recrystallized from a mixture of isopropyl alcohol and ethyl acetate. The yields of **BVK** and **QVK** were 37 and 43%, respectively.

### 2.3. Polymer Synthesis

For a radical bulk polymerization, monomer (**BVK**, **TVK**, or **QVK**) and initiator (AIBN, 0.05–4.0 mol % per monomer) were mixed in chloroform solution and poured into a glass ampule. After solvent vacuum evaporation, the ampule was filled with argon, sealed, and placed into an oil bath at 80 °C for the required time. Thermal self-initiated polymerization was conducted directly in the DSC cell. Thereafter, the 6–10 mg of the monomer was put into the aluminum pan, sealed, and placed in the DSC cell. After that, the heating from room temperature to 300 °C, with a desired heating rate in the range of 2.5–15 K/min, was started.

After the polymerization was finished, the weighed amount of the sample was dissolved in THF and analyzed by GPC after filtration through a 0.45 µm PTFE filter.

For solution polymerization, the required amounts of monomer, AIBN, and RAFT agent (DoPAT or CPDTC), where applicable, were dissolved in 1,4-dioxane and poured into an ampule. The solution was degassed by three freeze–pump–thaw cycles, and the ampule was sealed and placed into an oil bath at 80 °C for the required period of time. After polymerization, the sample was cooled down to room temperature and the solvent was vacuum evaporated. Products were dissolved in THF and analyzed by size exclusion chromatography (SEC) after filtration through a 0.45 µm PTFE filter.

### 2.4. Characterization

Molecular weight characteristics of the polymer were obtained by SEC on a high-pressure modular liquid chromatograph equipped with a Lab Alliance Series 1500 Constant Flow Pump (Scientific Systems, USA), a Refractive Index Detector 2142 (LKB Bromma, Switzerland), and columns Waters WAT054460 and TOSOH Biosep G3000HHR. THF was used as the eluent. The chromatograms were processed using the MultiChrom (Ampersand, Russia) software package. Molecular weights were calculated relative to polystyrene standards.

The ^1^H and ^13^C NMR spectra of the monomers were recorded on an AvanceTM 600 spectrometer (Bruker, Germany). The operating frequency for protons and ^13^C nuclei was 400 and 101 MHz, respectively. The chemical shifts of protons and ^13^C nuclei were determined relative to the residual signal of chloroform (7.26 ppm) or the signal of CDCl_3_ (77 ppm), respectively, and converted to TMS. The accuracy of determining chemical shifts was no worse than 0.001 ppm and 0.03 ppm, respectively.

The optical textures and thermal transition temperatures were evaluated using polarising optical microscope (POM) Polam L-213 (LOMO, Russia) equipped with a FP-82 HT hot stage (Mettler Toledo, Switzerland) and a temperature control unit.

DSC measurements were carried out with Mettler DSC 823E (Mettler Toledo, Switzerland) using 40 μL Al pans at a heating rate of 10 K·min^−1^ under an argon atmosphere. Melting points and enthalpies of indium and zinc were used for temperature and heat capacity calibration.

IR spectra were recorded in the reflection mode on an IFS-66v/s Bruker FT-IR spectrometer using an ATR attachment (scan 50, ZnSe crystal, resolution 2 cm^−1^, range 600–4000 cm^−1^). The spectra were processed using the OPUS-7 software package.

## 3. Results and Discussion

### 3.1. Synthesis of Monomers and Their Phase Behavior

The synthetic route for obtaining the desired vinyl ketones **BVK** and **QVK** includes the following stages:-the synthesis of the corresponding cyclopropan-1-ols from the esters using the Kulinkovich reaction;-cycle splitting followed by its transformation to the corresponding substituted 2-bromoethyl ketone;-dehydrobromination of 2-bromoethyl ketone and isolation of target vinyl ketone (see Figure 1 and Figure 2).

All the intermediate compounds have been isolated in high yields and the targeted vinyl ketones have been purified and their structure was confirmed by NMR and FTIR spectroscopy. The characteristic part of ^1^H NMR spectra of BVK and QVK is given in Figure 5. The original ^1^H, ^13^C NMR, and FTIR spectra of both monomers are given in Appendix A, and the assignment of the NMR signals is given in Table 1. Both spectra for **TVK** are given in our previous work [38]. The NMR spectra of **BVK** and **QVK** are in accordance with theoretical values of chemical shifts and confirm the structure of the synthesized monomers.

All the synthesized aryl vinyl ketones contain bulky aromatic substituents linked to soft methylene units. It may be supposed that these monomers may exhibit liquid crystalline behavior. To verify it, we analyzed their DSC thermograms and optical textures obtained by means of POM.

BVK containing diphenyl moieties forms crystals, which melted at 63 °C into an isotropic phase. This transition was accompanied by the melting enthalpy of 122 J/g (Figure 6a, curve 1). The crystal—Sm transition in TVK containing terphenyl moieties was characterized by the melting point at 55 °C and ΔH equal to 50.3 J/g (Figure 6a, curve 2) as we have shown previously [39]. The Sm—N transition effect was hardly observable on the first heat DSC thermogram, but it appears as a minor peak at 221 °C during the second heat (Figure 6b, curve 2). The isotropic transition was going on after 285 °C. As for the quaterphenyl vinyl ketone QVK containing tetraphenyl moieties, its DSC curve was more complex and was characterized by different phase transitions including melting to, presumably, smectic A phase at 33 °C and isotropization at 199 °C (Figure 6a, curve 3).

First, the heating of all the monomers results in the appearance of the broad exothermic peak on the DSC curves (Figure 6a, arrows). However, this peak disappears completely during further cooling and second heating and does not appear for corresponding polymers (Figure 6b). Meanwhile, the locus and the shape of other peaks also change. Thus, it may be supposed that the exothermic peak dealt with the chemical reactions of the monomer.

Polarizing microscopy of vinyl ketone monomers complements the DSC data. Transformation of the crystalline texture of BVK into isotropic one occurs without mesophase formation (Figure 7a–c). TVK forms a liquid crystal phase above the melting point which was characterized by the isotropic texture (Figure 7e). However, with further heating of TVK, we observed an active liquid movement, due to which the black non-birefringent phase becomes colored and birefringent at the moment (Figure 7f). This behavior indicates that the observed system does not correspond to an isotropic liquid, but retains a certain order in the arrangement of the long axes of the molecules, which are oriented at rest perpendicular to the observer. As the liquid moves, the axes of the molecules tilt, making the phase visible in crossed polarizers (Figure 7f). Therefore, the isotropic texture (Figure 7e) corresponds to the homeotropic LC phase. With a further increase in temperature, a mixed thread-like/schlieren nematic texture appears (Figure 7g), followed by the isotropic transition. QVK melts into a liquid crystal phase characterized by a fan-shaped texture (Figure 7i) at the beginning and a bâttonet-like texture at higher temperatures. Both of these textures are typical for the Smectic A phase. As the isotropic melt texture at higher temperatures represents a homogeneously dark image, this texture (Figure 7c) is common for all three monomers. Summarizing, one can conclude that, contrary to BVK, two other vinyl ketones form a liquid crystalline phase in a broad temperature range.

### 3.2. Synthesis of Polymers and Their Phase Behavior

As we have shown earlier, the formation of oligomers occurs during the heating of **TVK** in DSC cells up to 240–300 °C with M_n_ ~4 kg·mol^−1^ and Đ = 1.2–1.3 [38]. Meanwhile, the resulting product was characterized by the disappearance of the stretching band ν_C=C_ at 1677 cm^−1^ confirming the oligomerization of **TVK**. The products obtained after heating of **BVK**, **TVK**, and **QVK** in DSC cells with yields of 80–90% were analyzed by means of SEC. All of them are oligomers with Mn = 3–4 kg·mol^−1^ and Đ = 1.3 for **TVK** and **QVK** and 3.3 for BVK. Thus, the observed exotherms (Figure 6a) may be due to the self-polymerization of corresponding vinyl ketones. The onset temperature of self-polymerization was equal to 82, 163, and 213 °C for **BVK**, **TVK**, and **QVK** increasing with the growth of the size of the aromatic core.

The structure of **BVK** and **QVK** oligomers formed upon heating in the DSC cell was confirmed by FTIR spectroscopy. FTIR spectra of monomers were in good agreement with BVK and QVK structures. The characteristic bands are shown in Appendix A. The stretching vibrations bands at 1697 and 1678 cm^−1^ and bending vibrations at 1011, 992, and 968 cm^−1^ characteristic for the vinyl ketone group were present in both spectra although there was a difference in the relative band intensities.

Polymerization of **BVK** and **QVK** was accompanied by the opening of double C=C bonds in both monomers. Thus, the absence of the absorbance corresponding to stretching and bending vibrations of the vinyl group at about 1664 and 977 cm^−1^ in the FTIR spectra of the products (Figure 8) proves the completeness of the polymerization. Besides, a conjugation that forms in the monomer between C=C and C=O bonds ceases to be in the oligomer, resulting in a sharp decrease in the intensity of the band of stretching vibrations of the C=O group and its shift to the region of short waves from 1697 (**BVK**) to 1708 cm^−1^ (**polyBVK**) and from 1700 (**QVK**) to 1712 cm^−1^ (**polyQVK**). The bands characterizing the structure of other functional groups were present both in the spectra of the monomer (Appendix A) and the corresponding oligomer with a slight shift and a change in relative intensity. Stretching and bending vibrations of alkyl (-CH_2_-, -CH_3_) groups were observed at 2926, 2859, 1488, 1467, 1382, and 722 cm^−1^. Stretching and bending vibrations in 1,4- and 1,2,4-substituted aromatic rings could be seen at 3028, 1605, 1585, 1521, 812, and 742 cm^−1^, vibrations of C–O bonds were revealed at 1249, 1178, 1106, and 1043 cm^−1^.

Based on these results, we have studied the polymerization of the synthesized vinyl ketones at a constant temperature. When the common radical initiator AIBN was added to the vinyl ketone, radical polymerization proceeds upon heating at 80 °C. This radical mechanism was confirmed by the influence of AIBN concentration on both the polymerization rate and average molecular weights of the polymers formed. For example, in the case of **BVK** at AIBN concentration 0.05 mol % with respect to monomer, 5 days were required to reach 68% of monomer conversion, while at 0.4 mol % of AIBN the same conversion was reached after 17 h. The rise of AIBN concentration leads to a decrease in MW (Figure 9, curves 1 and 2, Table 2). The conventional radical polymerization of **BVK**, either in bulk or in 1,4-dioxane, solution leads to the formation of **poly(BVK)** with broad MWD (Figure 9). The increase in the number of phenyl substituents in vinyl ketone structure was accomplished by a decrease in monomer conversion and narrowing of MWD. The addition of the RAFT agent results in a simultaneous decrease in monomer conversion and MWD. Notably, the use of the RAFT technique provides the formation of polyvinyl ketone with low dispersity. Both RAFT agents provide good control over MW and dispersity confirming the high efficiency of leaving groups for **BVK** and **TVK** polymerization.

One of the attractive properties of polyvinyl ketones is their ability to undergo photodegradation after exposure to UV light. Taking **poly(BVK)** as an example, we have tested its resistance to UV light. **Poly(BVK)** was dissolved in THF at concentrations of 1.25 and 2.5 mg/mL. The prepared solution was irradiated with the light with λ = 365 nm and irradiation power of 1 W/cm^2^ for desired periods of time. Figure 10a illustrates the SEC curves of the initial polymer and polymer after irradiation for 40 h. As is seen, the SEC curve shifts to the region of lower MWs. The time-dependence of M_n_ is presented in Figure 10b. Increasing irradiation time leads to a noticeable decrease in the MW of the polymer. As is seen, the time-dependence of M_n_ is similar for both polymer concentrations and is described by linear dependence at irradiation time below 48 h, and, after that, the deviation from the linearity is observed. These features are similar to first-order kinetics, but the detailed study of the photo-degradation mechanism is beyond this research. Thus, taking **poly(BVK)** as an example, we have confirmed that new polyvinyl ketones containing bulky substituents are photodegradable.

Since polymers are derived from the mesogen-containing monomers, the LC properties of PVKs were also investigated. According to polarized microcopy, **TVK** and **QVK** in polymer form have LC phases. Schlieren texture (Figure 11b) becomes the only observable texture for the nematic LC phase of **TVK** and thread-like texture does not appear in the polymer form (see also Appendix A). Focal conic smectic A texture of the polymer **QVK** (Figure 11c) replaces the fan-shaped texture of the corresponding monomer. It is worth noting that having just crystalline **BVK** after polymerization indicates the presence of an LC form although of a non-characteristic texture (Figure 11a,d). An important feature of the polymer backbone is the ability to save liquid crystalline ordering upon cooling down. The focal conic texture of **QVK** was saved in solid polymer film with minor alteration (Figure 11f). Textures in higher resolution are added in Appendix A.

## 4. Conclusions

In the present research, we have synthesized new vinyl ketone monomers and their based polymers. These monomers have similar (phenyl-4-oxy)oct-1-en-3-one moiety and differ by the nature of the substituent attached to the para-position of the phenoxy group. All the synthesized monomers have bulky substituents and are able to form a crystalline phase. The ability to exhibit liquid crystalline behavior depends on the number of phenyl substituents attached to the para-position of the phenoxy group. The **BVK** with a biphenyl structure was unable to form a liquid crystalline structure and melts at 63 °C, while **TVK** and **QVK** having ter- and tetraphenyl moieties form different LC structures.

It is essential that all the monomers are able to self-polymerize upon heating and to produce LC polymers. The study of radical polymerization of **BVK**, **TVK**, and **QVK** reveals that all of them can undergo free radical polymerization in bulk or in solution with AIBN as the initiator. RAFT polymerization of the synthesized vinyl ketones using asymmetrical trithiocarbonates was successfully applied to **BVK** and **TVK** yielding oligomers with narrow MWD. The photo-degradation ability of the synthesized poly(vinyl ketones) was proven for **BVK** polymer under UV irradiation.

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
