# Peer review of "Novel Mesogenic Vinyl Ketone Monomers and Their Based Polymers"

_polymers, 2022, doi:10.3390/polym15010005_

Round 1
Reviewer 1 Report
The manuscript reports synthesis and characterization of two newly vinyl ketone monomers and their corresponding polymers. Photo-degradation of BVK polymer was obtained under UV light irradiation. The results can be accepted after considering the following possible revisions.
1) There are some impurities from NMR of Figure 5a,b.
2) To identify the kind of LC, SAXS of samples are necessary.
3) The yield of the final product should be provided in the synthesis.
Author Response
To reviewer 1
Dear reviewer,
First of all, we would like to express our thanks for your comments and suggestions. Please find below out response.
- There are some impurities from NMR of Figure 5a,b (old version)
You are absolutely right. The impurities represent the Br-compounds (5 and 5a in Schemes 1 1nd 2) (new version) because its Rf values are very close to each other and several recrystallizations and separations do not help. However the impurities do not disturb the polymerization process and do not appear in the product.
We have presented the characteristic parts of 1H NMR spectra in Figure 5 a. b. The original 1H, 13C NMR and FTIR spectra of both monomers are given in Figures S1–S3 in the Supplement.
The identification of 13C spectra is given in Table 1 with the accuracy provided by this method.
The corresponding text “The characteristic part of 1H NMR spectra of BVK and QVK is given in Figure 5. The original 1H, 13C NMR and FTIR spectra of both monomers are given in Figures S1–S3 “ is placed in
Page 6 of the new version.
- To identify the kind of LC, SAXS of samples are necessary.
You are again right. However for most liquid crystals having pretty simple textures their use for identification is enough. We are working on the new paper regarding the SAXS measurements of new monomers (and polymers) and hope to publish it is the nearest future.
- The yield of the final product should be provided in the synthesis.
We have added the final yields of BVK and QVK as 37 % and 43 %. These data are given in p.5 (new version)/
Reviewer 2 Report
This manuscript described the synthesis of two similar vinyl ketones monomers (BVK, QVK) and their based polymers. The authors further demonstrated the phase behaviors of these monomers and polymers. The work is original and well organized. However, the reviewer still has several concerns about the present manuscript.
1. Fig.5 is not clearly for reading. It is suggested to indicate the carbon atom corresponding to each peak on the 13C spectra. Some suggestion for Fig.6.
2. Original FTIR spectra is suggested to demonstrate.
3. Figures should be described in order. It is not convenient for the reviewer to read Fig.7 and Fig.8, largely because the descriptions are not appeared as the figure order.
4. Contents of Line 243-248 the explanation of the results in Fig. 7. Therefore, this part is suggested to appear in 3.1.
5. The temperature unit of °C is not correctly demonstrated, such as in Line 241. Please make a double check.
6. Why only one image of Fig 8c represents three results of BVK at 70 °C; TVK 300 °C and QVK at 300 °C?
7. In Fig.9, there are no a and b in the image, but 1 and 2.
8. Commonly, Fig 7, 9, 10 and 11 should have vertical coordinate axis.
Author Response
Dear reviewer, we would like to express our thanks for your comments and suggestions.
- 5 is not clearly for reading. It is suggested to indicate the carbon atom corresponding to each peak on the 13C spectra. Some suggestion for Fig.6. In accordance with the suggestions we have divided Fig. 5 in two figures. First part of it is given in Ffigure 5 of the new version and the second part which contains the original 1H, 13C NMR and FTIR spectra of both monomers summarizes Figures S1–S3 “ in the Supporting file.
- Original FTIR spectra is suggested to demonstrate
It is done in Figure S3 in Support.
- Figures should be described in order. It is not convenient for the reviewer to read Fig.7 and Fig.8, largely because the descriptions are not appeared as the figure order.
- Contents of Line 243-248 the explanation of the results in Fig. 7. Therefore, this part is suggested to appear in 3.1.
We have changed the numbers of figures. DSC curves are placed under number 6 in pp. 8, 9 and its description is located above the figure. As for POM images, they are summarized in Figure 7 and the description is given in p.10.
Both paragraphs and figures are shifted down to part 3/1
- The temperature unit of °C is not correctly demonstrated, such as in Line 241. Please make a double check
It is corrected in the new version.
- Why only one image of Fig 8c represents three results of BVK at 70 °C; TVK 300 °C and QVK at 300 °C?
As the isotropic melt texture at higher temperatures represents homogeneously dark image, this texture (Figure 7c, new version)) is common for all three monomers. That is why images of all three samples are summarized in one figure. This info is added in the paper. Another goal is to save the number of figures presented in the paper.
- In Fig.9, there are no a and b in the image, but 1 and 2.
Now it is Figure 8 and we have corrected the numbering.
- Commonly, Fig 7, 9, 10 and 11 should have vertical coordinate axis (here old numbers from the first version are given)..
We have added vertical axes to all the necessary spectra.